# Illness Among Mental Health Service Workers and Its Repercussions: A Scoping Review

**DOI:** 10.3390/ijerph22121822

**Published:** 2025-12-05

**Authors:** José Mateus Bezerra da Graça, José Adelmo da Silva Filho, Eslia Maria Nunes Pinheiro, Rosiane Davina da Silva, José Benedito dos Santos Batista Neto, Elisângela Franco de Oliveira Cavalcante

**Affiliations:** 1Department of Collective Health, Universidade Federal do Rio Grande do Norte, Natal 59078-970, RN, Brazil; eslianunes@gmail.com (E.M.N.P.); rosiane.uepb@outlook.com (R.D.d.S.); elisangelafranco2@gmail.com (E.F.d.O.C.); 2Department of Nursing, Universidade de São Paulo, São Paulo 05508-900, SP, Brazil; adelmof12@gmail.com; 3Department of Nursing, Universidade Federal do Pará, Belém 66075-110, PA, Brazil; netto1443@gmail.com

**Keywords:** mental health, nursing, mental health services, health professionals, occupational health

## Abstract

Various structural, organizational, and subjective factors contribute to psychological distress, exacerbated by adverse working conditions in mental health services and resulting in significant impacts on workers’ health. This study aims to map the existing literature on work-related factors and consequences associated with the illness of mental health service workers. A scoping review was conducted following the PRISMA-ScR guidelines. The review question was formulated based on the population, concept, and context framework. Searches were performed in eight electronic databases, and the findings were synthesized in synoptic tables. A total of 28 studies were included, addressing factors encompassing structural, organizational, and procedural aspects of the work environment, as well as social and relational elements involving the healthcare team, users, and their families, all contributing to workers’ illness. The main challenges identified include work overload, excessive working hours, inadequate physical infrastructure, insufficient supplies, understaffing, lack of managerial support, and exposure to physical, biological, chemical, and ergonomic risks. The analysis of work-related factors in mental health services reveals a concerning scenario of physical, emotional, and mental strain among professionals.

## 1. Introduction

The social precariousness of work generates complex dynamics that manifest in precarious working and employment conditions, characterized by persistent feelings of insecurity and vulnerability among workers, particularly related to the risk of unemployment. This situation is driven by multiple factors, including fragile forms of entry into the labor market, social inequalities, increased work intensity, outsourcing, lack of job security, and multiple occupational exposures to health risks. Long working hours, lack of autonomy, task repetitiveness, and monotony also contribute to such precariousness [1,2].

These transformations in the labor landscape have led to significant changes in the healthcare field, especially concerning production processes, professional profiles, and working conditions. The changes brought about by the Brazilian Psychiatric Reform (BPR) sought to overcome the precariousness of work in mental healthcare settings. However, recent setbacks and counter-reforms have introduced new challenges and impacts on workers [3,4].

In this context, mental healthcare has come to encompass not only technical and procedural aspects but also emotional, ethical, ideological, and political dimensions. Thus, in daily practice, healthcare workers and professionals involved in mental healthcare face a transition from an institutionalized model to a psychosocial model [3,4]. Among these professionals, the nursing staff is the most extensively involved, traversing all aspects of this transformation scenario and its implications for mental healthcare. In Brazil, out of 3.5 million health sector workers, approximately 1.7 million (around 50%) are nursing professionals. This category maintains the closest contact with patients in care processes, reinforcing the notion of transversal engagement, which consequently increases susceptibility to mental health disorders [5,6].

Therefore, factors related to organizational conditions, job demands, and the competencies of workers and professionals in mental health services—as well as cultural, social, and personal characteristics embedded in the work environment—are health determinants that can negatively affect well-being. Within this context, a literal distinction arises between ‘workers’ and ‘professionals’: while all professionals are workers performing duties in a specific area of expertise, usually attained through formal education, training, and practical experience, the term ‘worker’ is broader and includes anyone engaged in labor activity, regardless of their level of specialization or formality [7].

Timely monitoring and interventions regarding these work characteristics are essential for creating conditions that support the development of a satisfactory and fulfilling occupational identity and performance [8]. Epidemiological studies in the field of occupational health have highlighted the correlation between work organization characteristics and mental disorders [9]. Regarding workers and professionals in mental health services, significant gaps remain in both understanding the problem and identifying and addressing the factors contributing to these conditions. Workers affected by Work-Related Mental Disorders (WRMDs) may experience irritability, insomnia, fatigue, memory lapses, difficulty concentrating, and reduced physical and intellectual performance, as well as somatic complaints, discouragement, anxiety, depersonalization, and decreased productivity and job satisfaction [8,10,11,12].

From a perspective aimed at understanding the workers’ dynamics and the feelings of suffering and pleasure derived from the work environment, two main categories can be identified: the organizational structure, working conditions, and relationships; and the worker’s subjective engagement, including defensive strategies and spaces for collective discussion. These elements are directly related to experiences of pleasure and discomfort at work. The relationship between job demands and defense mechanisms against their psychological effects are crucial points in the psychodynamic approach to work, highlighting labor organization as a potentially destabilizing factor for the mental health of workers, including those in mental health services [13,14].

It is widely acknowledged that the work environment has a direct impact on workers’ health. Changes in the organization and processes of work have contributed to increased rates of illness among health professionals in recent years [10,11]. From this perspective, research can expand and update scientific knowledge on the subject, providing essential data for the implementation of occupational health policies—particularly for nursing workers—with a focus on improving working conditions, reducing exposure to environmental and psychosocial hazards, and ensuring protective measures for those working in mental health services. Accordingly, this review aims to map the existing literature on work-related factors and consequences that contribute to the illness of workers in mental health services.

## 2. Materials and Methods

This is a scoping review study, for which ethical approval by a Research Ethics Committee involving human participants was not required. This study was conducted as a scoping review rather than a systematic review. Scoping reviews are designed to map the existing literature, summarize key findings, and identify knowledge gaps, whereas systematic reviews focus on answering specific research questions and often evaluate the effectiveness of interventions. Given the exploratory nature of our study and the diversity of study designs, populations, and outcomes, a scoping review was considered the most appropriate method to provide a comprehensive overview and guide future research.

The aim was to gather the existing literature on the topic, summarize the data, and identify gaps in the current body of knowledge. To guide and support this study, the assumptions of the Joanna Briggs Institute (JBI) Reviewer’s Manual and the recommendations of the Preferred Reporting Items for Systematic Reviews and Meta-Analyses extension for Scoping Reviews (PRISMA-ScR): Checklist and Explanation were adopted, see Appendix A. The review protocol was registered on the Open Science Framework [15,16].

The review question was as follows: What work-related factors contribute to the illness of workers in mental health services? The PCC acronym was used (P: Population = workers in mental health services; C: Concept/phenomenon of interest = work-related factors that cause illness; and C: Context = mental health services).

The search strategy was carried out in three stages: an initial limited search in Medline and the Virtual Health Library (VHL) to identify the main concepts related to the research topic and apply them to the research question; listing of terms/descriptors to be used in the search strategy, with predefined use of Boolean operators such as AND and OR; and a manual search of the references from selected articles to identify studies not retrieved through the initial database strategy.

Searches were conducted in the following electronic databases: National Library of Medicine (PubMed); Cumulative Index to Nursing and Allied Health Literature (CINAHL); Virtual Health Library (VHL); Web of Science (WOS); Scientific Electronic Library Online (SciELO); Latin American and Caribbean Health Sciences Literature (LILACS); and Embase.

For gray literature, the Brazilian Digital Library of Theses and Dissertations (BDTD) was consulted. Table 1 presents the descriptors used in developing the search strategy, which was initially created in PubMed and later adapted to the indexing structure of each database.

The searches were conducted between 25 February and 29 April 2024, without language restrictions; however, the majority of the studies identified were in English. Studies in other languages, such as Spanish and Portuguese, were also included when titles and abstracts were understandable in English or translated using automated translation tools. No studies in Japanese, Korean, or other languages outside English, Portuguese, or Spanish were identified in the final selection. Of the 28 studies included, 22 were in English, 4 in Portuguese, and 2 in Spanish.

Database searches were carried out simultaneously by two researchers, who followed the same order of descriptor use and Boolean combinations across each database and subsequently compared their results. To ensure comprehensive coverage, all databases were accessed via the Coordination for the Improvement of Higher Education Personnel (CAPES) journal portal.

The following types of studies were considered eligible: opinion articles, reviews, case studies, quasi-experimental studies, randomized clinical trials, cohort studies, case–control studies, cross-sectional studies, and qualitative studies that addressed work-related factors and consequences contributing to the illness of workers in mental health services. Exclusion criteria included incomplete documents, duplicates, and studies that did not present work-related factors potentially associated with worker illness.

After the searches were completed in both databases and gray literature sources, duplicate articles were removed using Rayyan® software, version 1.0 (Cambridge, MA, USA). Next, titles and abstracts were screened based on the established eligibility criteria. Studies deemed suitable were included for full-text review and considered for the final selection of the review literature.

In the event of disagreements between reviewers, a third reviewer was consulted to resolve the conflict. Data extraction was conducted using an Excel spreadsheet to organize the most relevant findings for structuring and grouping the literature. Extracted data included: Authors; Year/Country; Method/Design; Objective; Population; Setting; Related factors; and Consequences.

Data were presented both textually and visually through a narrative synthesis, followed by a discussion contextualizing the findings in relation to the study’s specific objectives. These objectives were to identify work-related factors contributing to the illness of mental health professionals, categorize the associated consequences, and map gaps in the current literature to address the guiding research question.

## 3. Results

The search process initially identified 1748 records, of which 754 were duplicates. A total of 1030 articles were included for initial screening. Following a second screening, 431 records were excluded due to lack of full-text availability, and 512 were excluded based on title and abstract analysis. Of the remaining 87 articles, 28 met the eligibility criteria and addressed the proposed objectives, and were thus included in the final sample of this review (Figure 1).

The studies were conducted in different years and countries, with a significant predominance of research originating from Brazil [8,12,17,18,19,20,21,22,23,24,25,26,27,28,29,30,31,32,33,34,35,36]. Regarding the years of publication, they range from 2006 to 2024, including studies carried out in Switzerland [37], Israel [38,39], Iran [40,41], and Belgium [42]. The study settings described in the articles encompass various types of mental health services, covering a range of contexts and regions. Notably, multiple types and locations of Psychosocial Care Centers (CAPS) are represented, including CAPS I, II, III, AD, and CAPSi, from cities in Brazil’s northeast—such as Campina Grande, in the state of Paraíba—to the south, such as Foz do Iguaçu, in the state of Paraná. In addition, the studies include psychiatric inpatient units in public and private hospitals [26,28,30,33,34,37,40,41,42], psychiatric teaching hospitals [31], mental health centers in Israel [38,39], Iran [41,42], and Belgium [42], as well as specialized services for the care of individuals who use alcohol and other drugs [17,35] (Table 2).

The predominance of Brazilian studies in this review reflects the relevance and significant scientific production in the field of mental health in Brazil, particularly after the psychiatric reform, which stimulated research on working conditions and the health of professionals in community-based services such as CAPS. International publications were included to broaden the comparative analysis; however, their number was smaller due to the specificity of the topic in the Brazilian context.

Regarding the study populations, participants included physicians, nurses, nursing assistants, psychologists, social workers, occupational therapists, administrative staff, physical educators, pharmacists, physiotherapists, speech therapists, caregivers, security guards, and general services workers. Sample sizes varied considerably, ranging from small groups of 6 workers to large studies involving up to 4464 professionals. In terms of study designs, quantitative investigations were more predominant, particularly those with cross-sectional designs [8,19,21,28,29,33,34,37,38,39,40,42]. Qualitative studies were also prominent, employing exploratory and descriptive approaches [17,18,20,22,23,24,25,26,30,31,36]. Additionally, some studies utilized mixed-methods approaches [12,32,35] and reviews of the literature [27].

Based on the interpretative analysis of Figure 2, it is possible to characterize the work-related factors that contribute to the illness of workers in mental health services, as well as their possible consequences. Work overload was identified as a critical factor affecting these workers, further exacerbated by the COVID-19 pandemic. The constant pressure to meet targets and the excessive workload [8,9,10,11,12,13,14,15,16,17,18,19,20,21,22,23,24,25,26,27,28,29,30,31,32,33] resulted in fatigue and psychological distress [12,13,14,15,16,17,18,19,20,21,22,23,24,25,26,27,28,29,30,31,32,33,34,35,36], stress [8,9,10,11,12,13,14,15,16,17,18,19,20,21,22,23,24,25,26,27,28,29,30,31,32,33,34,35,36], sleep disturbances [8,9,10,11,12,13,14,15,16,17,18,19,20,21,22,23], and Burnout syndrome [12,13,14,15,16,17,18,19,20,21,22,23,24,25,26,27,28,29,30,31,32,33,34,35], with job dissatisfaction as a major consequence [12,13,14,15,16,17,18,19,20,21,22,23,24,25,26,27,28,29,30,31,32,33,34,35,36,37].

Inadequate physical infrastructure and irregularities in facilities [8,9,10,11,12,13,14,15,16,17,18,19,20,21,22,23,24,25,26,27,28,29,30,31,32,33,34,35,36], combined with a shortage of supplies and materials [12,13,14,15,16,17,18,19,20,22,26,27,31,33,36], were described as having a negative impact on the quality of the work environment. Work organization, understaffing [12,13,14,15,16,17,18,19,20,21,22,23,24,25,26,27,28,29,30,31,32,33,34,35,36,37,38,39,40,41,42,43,44,45], lack of managerial support [23,24,25,26,27,28,29,30,31,32], and insufficient clinical supervision [26,27,28,29,30,31,32,33,34,35,36,37,38,39,40,41] contribute to feelings of insecurity [12,13,14,15,16,17,18,19,20,21,22,23,24,25,26,27,28,29,30,31,32,33,34] and irritability [23,24,25,26,27,28,29,30] among professionals. These conditions negatively affect both overall and mental health [20,21,22,23,24,25,26,27,28,29,30,31,32,33,34,35,36], as well as interpersonal relationships in the workplace, and even personal and family life [23].

Physical demands [35,36,37] and excessive work pace [12,13,14,15,16,17,18,19,20,21,22,23] were also evident in the findings, resulting in emotional exhaustion [21,22,23,24,25,26,27,28,29,30,31,32], depersonalization [21,22,23,24,25,26,27,28,29,30,31,32,33,34,35,36,37,38,39,40,41], low professional accomplishment [21], sadness [12,23], feelings of frustration [8,12,18], failure [18,19,20,21,22,23,24,25,26,27,28,29,30,31,32,33,34,35,36,37,38,39,40,41], and powerlessness [18,19,20,21,22,23,24,25,26,27,28,29,30,31,32,33,34,35,36,37]. Interpersonal relationship problems [12,13,14,15,16,17,18,19,20,21,22,23,24,25,26,27,28,29,30,31,32,33,34,35], such as lack of team commitment [8], lead to fatigue [8,9,10,11,12,13,14,15,16,17,18,19,20,21,22,23,24,25,26,27,28,29,30,31,32,33,34], depressive symptoms [8,9,10,11,12,13,14,15,16,17,18,19,20,21], mood swings [23], shame [38], and feelings of guilt [36].

Physical risks (e.g., noise) [28], biological risks (e.g., bacteria and viruses) [28,29,30], chemical risks (e.g., tobacco) [28,29,30,31,32,33,34,35], and ergonomic risks (e.g., poor posture) [28,29,30] were prevalent. In addition, psychosocial risks (e.g., stress and physical aggression) [28,29,30,31,32,33,34,35] and precarious working conditions [24,25,26,27,28,29,30,31] contributed to hypertension [21], dermatological diseases [21], urinary tract infections [21], body, head, and back pain [12,13,14,15,16,17,18,19,20,21,22,23,24,25,26,27,28,29,30], fear [12,13,14,15,16,17,18,19,20,21,22,23,24,25,26,27,28,29,30,31], absenteeism [22], and fatigue [23].

Insufficient academic training [31] and lack of professional development opportunities [31], combined with the predominance of biomedical care practices [12,13,14,15,16,17,18,19,20,21,22,23,24,25,26], generate dissatisfaction and demotivation [23,25]. Stigmatization and prejudice toward service users [18,19,20,21,22], as well as a lack of understanding from users’ families [32], further intensify mental suffering [26], resulting in shame [38] and alcohol abuse [30].

Low wages [19,20,21,22,23,24,25,26,27,28,29,30,31,32,33,34,35,36] and lack of recognition [19,20,21,22,23,24,25,26,27,28,29,30,31,32,33,34,35,36] are additional factors that contribute to feelings of frustration [8,9,10,11,12,13,14,15,16,17,18] and fatigue [23], especially among professionals with longer tenure in the service [40].

## 4. Discussion

This study aimed to investigate the work-related factors that contribute to the illness of workers in mental health services, understanding that the health of healthcare workers requires reflective practices on the conditions and organization of work within these services. The workplace in mental health services can simultaneously function as a source of professional fulfillment and as a context for stress and suffering, depending on organizational and relational dynamics.

Although this study did not focus on a single professional category, the results highlighted nursing as the most frequently examined profession in research on workers’ mental health in mental health settings. The working conditions of nursing teams emerge as a critical issue in Brazil’s health system, characterized by an insufficient and inadequately trained workforce in mental healthcare settings—an issue closely related to nursing education in Brazil [31,43].

These problems negatively impact the health of nursing professionals, compromising the reception and care of users of health services, and affecting the quality and effectiveness of care delivery [44].

Despite the significant advances brought by the Brazilian psychiatric reform—such as the promotion of deinstitutionalization and the implementation of community- and outpatient-based actions—the country continues to face major challenges beyond workforce shortages. In recent years, the scenario has been marked by funding restrictions and the Unified Health System’s (SUS) difficulty in absorbing new demands, undermining the continued progress of the reform [45,46].

These structural and financial constraints have led to long working hours, insufficient rest, and continued pressure to meet high service demands without adequate support, which contributes to mental exhaustion, anxiety, and other mental health conditions. This creates a vicious cycle in which professionals become ill and leave the workforce, worsening the shortage of human resources—factors identified in this review that significantly affect the lives and health of mental health workers [19,20,21,22,23,24,25,26,27,28,29,30,31,32,33,34,35,36,37,38,39,40,41,42,43,44].

In this context, the findings suggest that organizational structures and understaffed teams directly affect the provision of comprehensive care to users and their families. The findings of study [47] support the idea that mental illness not only impacts healthcare workers themselves, but also the family and social context in which they are embedded. Therefore, the organizational structure requires teams to adopt a broad, integrated approach that includes collaboration with different components of the healthcare network, through mental health matrix support and clinical supervision—identified in this review as an associated factor due to its lack of consistent implementation.

The physical infrastructure of psychosocial care in Brazil shows that many services were established in buildings without the necessary adaptations to meet the needs of users, families, and healthcare teams. In contrast, it is known that a healthcare facility that adheres to design principles of therapeutic environments becomes more welcoming, helping to strengthen and renew the patient’s relationship with the institution. These environments cease to be controlling and restrictive and instead promote conditions that enhance autonomy for individuals in their care processes [48].

Regarding workplace risks and safety, this review highlights studies that point to biological, chemical, and physical risks. Examples include biological risks associated with potential infections caused by bacteria, viruses, fungi, and parasites. These risks are present across mental healthcare settings, both community-based and hospital-based, and can compromise workers’ physical and psychological well-being [18,19,20,21,22,23,24,25,26,27,28,29,30,31,32,33,34,35,36,37,38,39,40,41,42,43,44,45,46,47,48,49].

The lack of an organizational culture focused on protecting workers in Brazilian mental health services is concerning. Without a system that prioritizes safety and well-being, work environments can become harmful, jeopardizing both the quality of services and workers’ health. Although there are laws and policies intended to protect workers, their implementation is often insufficient—due to lack of resources, weak enforcement, or a lack of genuine institutional commitment [31,43].

Studies show that long working hours reduce professional satisfaction across sectors and are linked to declining motivation over time. This can lead to personal dissatisfaction, which in turn undermines motivation and affects self-esteem. Some studies indicate that civil servants working in public institutions report greater satisfaction than workers in the private sector, likely due to the job security enjoyed by public employees compared to those on less secure contracts (e.g., temporary or outsourced workers) [19].

Mental health professionals’ job satisfaction is strongly influenced by the conditions in which services are delivered. A lack of alignment between service logistics and user needs was reported as a significant concern. Frequent interruptions during patient care, staff shortages, and delays in acquiring materials are problematic. These findings are supported by another study that also identified dissatisfaction due to service interruptions, resource scarcity, and organizational issues [18,19].

The implications for practice highlight the importance of an integrated approach to addressing the factors that contribute to illness among mental health service workers. Excessive workloads, performance pressure, and inadequate facilities must be strategically addressed to improve workplace conditions. Measures such as reviewing organizational processes, enhancing professional training, and improving physical and material resources are essential to mitigate the negative impacts identified.

Following the methodological framework of this scoping review, no statistical weighting or quantitative synthesis of prevalence or incidence rates was performed. Instead, to enhance interpretability, stressors were organized according to their recurrence and emphasis across the included studies. Primary stressors most frequently reported and consistently emphasized were work overload, understaffing, and long working hours. Secondary stressors that were recurrent but less prominent were inadequate infrastructure, exposure to occupational risks (physical, biological, chemical, and ergonomic), and insufficient supplies. Tertiary stressors less consistently highlighted or context-dependent were limited training opportunities and insufficient managerial support. This frequency-based categorization clarifies the relative salience of each factor in the literature while remaining aligned with the methodological boundaries of a scoping review.

Another relevant aspect concerns the terminology used to describe the workforce. In the Brazilian literature, the term worker (trabalhador) was often applied broadly to all individuals engaged in mental health services, regardless of their professional training or clinical responsibilities. In contrast, international studies more commonly used the term professional to designate licensed staff providing direct clinical care. This heterogeneity in descriptors limited the possibility of consistently distinguishing between clinically trained professionals (e.g., physicians, nurses, psychologists) and other service workers (e.g., administrative or support staff). While differences in educational background and role may influence how individuals experience and manage stressors, the available data did not allow for systematic stratification by these characteristics.

In addition to the findings discussed, this review identified several gaps in the literature. While factors such as workload, staff shortages, and exposure to physical and biological risks have been examined, there is limited research on the long-term psychological effects of work-related stress, differences in occupational risks across mental health service settings, and the effectiveness of interventions aimed at mitigating worker illness. Moreover, organizational culture, team dynamics, and socio-cultural factors remain underexplored, limiting understanding of how these elements influence the well-being of mental health professionals. Addressing these gaps through future research is essential to develop targeted strategies for improving workplace conditions, enhancing professional satisfaction, and ultimately safeguarding both workers’ health and the quality of care provided to users.

## 5. Conclusions

This study revealed that precarious working conditions in mental health services have a significant impact on the mental health of professionals and the quality of care provided. Excessive workloads, inadequate infrastructure, and lack of professional training are critical factors that must be addressed to improve the work environment and, consequently, the effectiveness of mental health services.

It is essential that occupational health policies consider mental health workers not only in structural and financial terms but also through the lens of an integrated approach that prioritizes worker safety and well-being. Strategic measures such as the revision of organizational processes, expansion of professional training, and improvement of physical and material conditions are crucial for mitigating the negative impacts identified and fostering a healthier and more productive work environment.

The study also highlights the importance of continuing the psychiatric reform and adapting work practices to new demands, ensuring that mental health services can effectively and humanely meet the needs of both professionals and service users.

## Figures and Tables

**Figure 1 ijerph-22-01822-f001:**
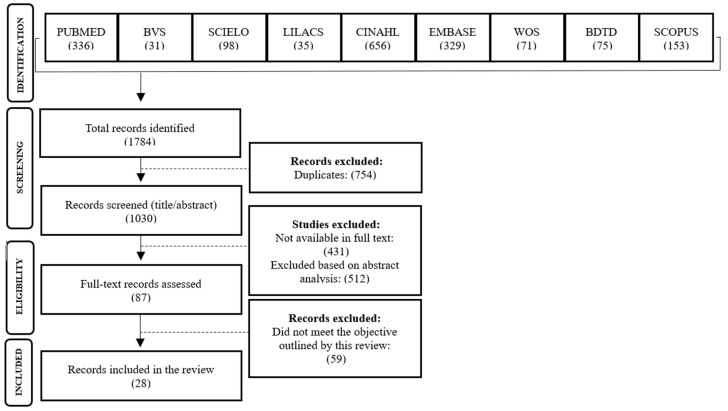
Flowchart of the study selection process, 2024.

**Figure 2 ijerph-22-01822-f002:**
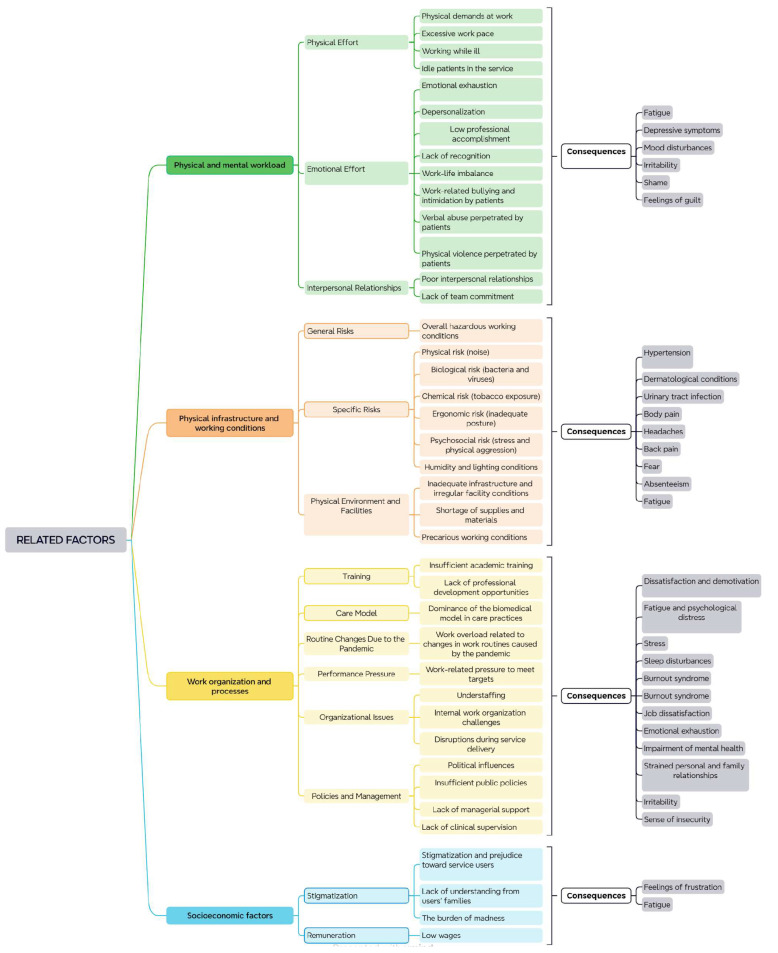
Work-related factors contributing to the illness of mental health service workers and their consequences, 2024.

**Table 1 ijerph-22-01822-t001:** Descriptors used in the search strategy for population, phenomenon of interest, and context, 2024.

Usage		Records Retrieved
1	‘Health Personnel’[MeSH Terms] OR ‘occupational groups’[MeSH Terms] OR ‘Work’[MeSH Terms] OR ‘Health Workers’[All Fields] OR ‘Health Workers’[All Fields] OR ‘Healthcare Workers’[All Fields] OR ‘Healthcare Providers’[All Fields] OR ‘Healthcare Worker’[All Fields] OR ‘Health Professional’[All Fields] OR ‘Health Professional’[All Fields] OR ‘health professionals’[All Fields] OR ‘health professionals’[All Fields]	893,481
2	‘Precipitating Factors’[MeSH Terms] OR ‘Causality’[MeSH Terms] OR ‘Risk Factors’[MeSH Terms] OR ‘Workplace’[MeSH Terms] OR ‘Workload’[MeSH Terms] OR ‘Working Conditions’[MeSH Terms] OR ‘Hazardous Work’[All Fields] OR ‘Precarious Work’[All Fields] OR ‘Occupational Stress’[MeSH Terms] OR ‘Occupational Health’[MeSH Terms] OR ‘Work-Life Balance’[MeSH Terms] OR ‘Occupational Diseases’[MeSH Terms] OR ‘Social Problems’[MeSH Terms] OR ‘Social Exploitation’[All Fields] OR ‘Worker safety’[All Fields] OR ‘Employee health’[All Fields] OR ‘Work-related stress’[All Fields] OR ‘Job-related stress’[All Fields] OR ‘Work Environment Stress’[All Fields] OR ‘Work Stress’[All Fields] OR ‘Workplace stress’[All Fields] OR ‘Bullying Work’[All Fields] OR ‘Work Environment’[All Fields] OR ((‘embarrass’[All Fields] OR ‘embarrassed’[All Fields] OR ‘embarrassing’[All Fields] OR ‘embarrassment’[MeSH Terms] OR ‘embarrassment’[All Fields]) AND (‘work’[MeSH Terms] OR ‘work’[All Fields])) OR ‘Abuse Work’[All Fields] OR ‘Incivility’[All Fields]	1,522,983
3	((‘Mental Health Services’[MeSH Terms] OR ‘hospitals, psychiatric’[MeSH Terms] OR ‘Psychiatric Hospital’[All Fields] OR ‘Mental Health Hospital’[All Fields] OR ‘Psychiatric Institute’[All Fields]) AND ‘Mental Health Institute’[All Fields]) OR ‘Psychiatric Center’[All Fields] OR ‘Mental Health Center’[All Fields] OR ‘Mental Hygiene Services’[All Fields] OR ‘Psychosocial Care Centers’[All Fields] OR ‘Community Mental Health Centers’[MeSH Terms] OR ‘CAPS’[All Fields]	37,585
1 AND 2 AND 3		336

Source: Authors, 2024.

**Table 2 ijerph-22-01822-t002:** Methodological Summary of Included Studies, 2024.

Authors	Year Country	Methods	Objective	Setting
Fidelis F.A.M. et al. [8]	2021 Brazil	Cross-sectional Study	To analyze the impact of workload on job satisfaction among CAPS workers in a municipality in the countryside of São Paulo.	Four services: CAPS I, CAPS AD, CAPS for children, and CAPS III.
Lima I.C.S. et al. [12]	2023 Brazil	Study Mixed-Methods	To analyze work precariousness through labor conditions that influence mental healthcare management and workers’ health.	Six CAPS.
Santos L.R. et al. [17]	2022 Brazil	Qualitative Study	To understand the life experiences of mental health workers during the first year of the coronavirus pandemic.	Two Psychosocial Care Centers (CAPS), one Psychiatric Inpatient Unit, and one Alcohol and Drug Referral Service.
Kolhs M. [18]	2019 Brazil	Qualitative Study	To explore the suffering and defense strategies of workers at a CAPS Alcohol and Drugs III, from the perspective of Dejours’ “Theatre of Work”.	CAPS AD III.
Oliveira J.F. et al. [19]	2019 Brazil	Cross-sectional Study	To evaluate job satisfaction and workload among nurses in mental health services and compare them with other professionals.	Eight CAPS in a municipality in the far south of Brazil.
Clementino F.S. et al. [20]	2018 Brazil	Qualitative Study	To assess service quality and professionals’ satisfaction regarding workload in CAPS in Campina Grande, Paraíba.	Five CAPS (two type I, one type II, one type III, and one AD).
Zanatta A.B. et al. [21]	2021 Brazil	Cross-sectional Study	To evaluate the prevalence of Burnout Syndrome among CAPS health professionals and associations with biosocial and occupational characteristics.	Eleven CAPS in a municipality in the interior of São Paulo State.
Bustamente V. et al. [22]	2022 Brazil	Qualitative Study	To examine work processes and their relationship with institutional suffering from René Käes’ perspective	Four Child and Youth CAPS (CAPSi).
Sousa Y.G. et al. [23]	2021 Brazil	Qualitative Study	To investigate work environment factors contributing to mental workload among nursing teams at CAPS III.	CAPSIII
Athayde V. et al. [24]	2011 Brazil	Qualitative Study	To analyze work situations in CAPS focusing on the health-illness-work process of healthcare professionals.	One CAPS in the northern region of Rio de Janeiro city.
Glanzener C.H. [25]	2008 Brazil	Qualitative Study	To assess sources of suffering in a CAPS team and identify strategies for coping.	CAPS II in Foz do Iguaçu, Paraná.
Magnus C.N. [26]	2009 Brazil	Qualitative Study	To understand the dynamics of the work process among mental health professionals in a public psychiatric hospital, focusing on suffering and pleasure.	State Psychiatric Hospital of Rio Grande do Sul.
Ramminger T. [27]	2009 Brazil	Literature Review	To collectively analyze the activities of CAPS workers.	Mental health services.
Fernande M.A. et al. [28]	2014 Brazil	Cross-sectional Study	To analyze the association between mental health workers’ illness and occupational risks.	Psychiatric hospital in Teresina, Piauí, Brazil.
Rebouça D. et al. [29]	2007 Brazil	Cross-sectional Study	To assess job satisfaction levels and their impact on mental health professionals, and identify associations with sociodemographic and functional variables.	Mental health institution for long-term care in Rio de Janeiro.
Carvalho M.B. et al. [30]	2006 Brazil	Qualitative Study	To identify stressors experienced by nursing staff in a psychiatric hospital, understand burnout processes, and analyze coping strategies that support quality of work life.	Private psychiatric hospital accredited by SUS in São Paulo.
Scozzafave M.C.S. et al. [31]	2019 Brazil	Qualitative Study	To characterize the presence of psychosocial risks related to nurses’ work in a psychiatric hospital and management strategies for these risks.	Medium-sized public psychiatric teaching hospital in São Paulo State.
Pascoal F.F.S. et al. [32]	2021 Brazil	Mixed-Methods Study	To identify factors causing work overload in a psychiatric complex and assess strategies to mitigate this overload.	Psychiatric Complex in João Pessoa, Paraíba.
Sousa K.H.J.F. et al. [33]	2018 Brazil	Cross-sectional Study	To analyze the risk of illness among nursing professionals in the context of a psychiatric hospital.	Psychiatric Hospital in Teresina, Piauí.
Vieira D.L.C. [34]	2017 Brazil	Cross-sectional Study	To characterize the sociodemographic profile of nursing technicians in psychiatric hospitals, and assess satisfaction and workload levels and their interrelations.	Two psychiatric hospitals in Minas Gerais, Brazil.
Souza I.A.S. et al. [35]	2015 Brazil	Mixed-Methods Study	To analyze the work process and its impact on nurses working in a mental health service for users of alcohol and other drugs.	Specialized mental health service for comprehensive care of users of alcohol and other drugs, São Paulo, Brazil.
Guimarães J.M.X. et al. [36]	2011 Brazil	Qualitative Study	To analyze satisfaction and/or dissatisfaction with work among mental health team members in CAPS in Fortaleza.	Three CAPS in Fortaleza, Ceará, Brazil
Peter K.A. et al. [37]	2024 Switzerland	Cross-sectional Study	To identify the extent of work-related stress and associated stressors, including cognitive and behavioral stress responses, burnout symptoms, health status, sleep quality, job satisfaction, and turnover intention.	Twelve psychiatric hospitals.
Assuline S.Z. et al. [38]	2022 Israel	Cross-sectional Study	To examine associations between exposure to social shame and patient bullying, perceived risk, burnout, professional functioning, and turnover intention among mental health professionals.	One Mental Health Center in Israel.
Itzhaki M. et al. [39]	2015 Israel	Cross-sectional Study	To explore the effects of exposure to violence, work-related stress, team resilience, and post-traumatic growth on life satisfaction among mental health nurses.	One Mental Health Center in Israel.
Ghasemi M.S. et al. [40]	2022 Iran	Cross-sectional Study	To investigate workload and burnout among nurses at a major public psychiatric hospital in Tehran, Iran.	Public Psychiatric Hospital in Tehran.
Ashtari Z. et al. [41]	2009 Iran	Study type not specified	To evaluate the relationship between professional performance and burnout among staff in a psychiatric hospital.	Raazy Psychiatric Center, Tehran, Iran.
Bogaert V.P. et al. [42]	2012 Belgium	Cross-sectional Study	To examine the relationship between nurses’ work environments, workload, burnout, work outcomes, and self-reported care quality in psychiatric hospitals.	Two public psychiatric hospitals in different regions of the Dutch-speaking part of Belgium.

Source: Authors, 2024.

## Data Availability

Not applicable.

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
