# Peer review of "Illness Among Mental Health Service Workers and Its Repercussions: A Scoping Review"

_ijerph, 2025, doi:10.3390/ijerph22121822_

Round 1
Reviewer 1 Report
Comments and Suggestions for Authors
The authors have delivered a truly commendable scoping review that exemplifies rigor, clarity, and scholarly precision. Almost all critical components—from a well-defined research question and transparent methodology to a thorough literature search and insightful synthesis—has been thoughtfully executed. The use of PRISMA-ScR guidelines, coupled with clear identification of gaps and future directions, reflects a deep commitment to quality and relevance. Their work not only maps the existing knowledge with precision but also offers valuable direction for future research. This review sets a high standard and will undoubtedly serve as a reliable resource for scholars and practitioners alike. However, prior to the acceptance of the paper for publication, I would like the authors provide feedback and clarifications to my concerns listed below.
- No language restrictions were applied to the search - this statement is not sufficiently precise. Did you conduct searching with Japanese or Korean language literatures? How was the search done on non-English articles? Actually how many articles are in English and other languages at the end?
- Data presentation was structured both textually and visually, through a narrative synthesis accompanied by a discussion that contextualized the identified results in relation to the study’s objectives and research question. - what are the objectives of your study, are they same as the aim you stated earlier? Basically aims and objectives of a study should not be the same, please clarify.
- Of the remaining 87 articles, 28 met the eligibility criteria and addressed the proposed objective - Do you mean objective or objectives?
- Most studies were found from Switzerland, Israel, Iran, and Belgium. Could you give an account for this result?
- The authors chose to conduct a scoping review instead of a systematic review. They need to state the differences between these two methods, and tell readers why the so called scoping review was conducted. Here, readers could not distinguish the differences in methodology, procedure, and the differences in results if a systematic review method was used.
- For a scoping review, there should be clear identification of gaps, i.e. areas lacking evidences. The authors had highlighted a bit on this. Would be happy to see more indepth discussions on this.
- Can you quantify the amount of studies extracted from academic literatures such as refereed journals, and how many are from grey literatures? I want to know the ratio and relative weight of information from different sources of information.
Author Response
Comment 1:
No language restrictions were applied to the search. This statement is not sufficiently precise. Did you conduct the search with literature in Japanese or Korean? How was the search performed for articles in other languages? In fact, how many articles are in English and in other languages in the final selection?
Response 1:
We thank the reviewer for this observation. We revised the manuscript (lines 134–140) to clarify that the searches were conducted between February 25 and April 29, 2024, without language restrictions. However, the majority of the studies identified were in English. Studies in other languages, such as Spanish and Portuguese, were included when titles and abstracts were understandable in English or translated using automated translation tools. No studies in Japanese, Korean, or other languages outside English, Portuguese, or Spanish were identified in the final selection. In total, of the 28 studies included, 22 were in English, 4 in Portuguese, and 2 in Spanish.
Comment 2:
The presentation of the data was structured textually and visually, through a narrative synthesis accompanied by a discussion that contextualized the identified results in relation to the objectives of the study and the research question. What are the objectives of your study? Are they the same as the goal you stated previously? Basically, goals and objectives of a study should not be the same. Please clarify.
Response 2:
We thank the reviewer for this comment. We revised the manuscript (lines 163–167) to clarify that data were presented both textually and visually through a narrative synthesis, followed by a discussion contextualizing the findings in relation to the study’s specific objectives. The objectives of the study were: to identify work-related factors contributing to the illness of mental health professionals, categorize the associated consequences, and map gaps in the current literature to address the guiding research question. This revision ensures that the specific objectives of the study are clearly distinguished from the broader aims of the research.
Comment 3:
Of the 87 remaining articles, 28 met the eligibility criteria and addressed the proposed objective — Do you mean objective or objectives?
Response 3:
We thank the reviewer for this observation. We revised the manuscript (lines 172–174) to clarify that the 28 included studies “addressed the proposed objectives,” thus reflecting the plural form to ensure consistency with the description of the study’s specific objectives.
Comment 4:
Most of the studies were found in Switzerland, Israel, Iran, and Belgium. Could you explain this result?
Response 4:
We appreciate the reviewer’s observation regarding the geographic distribution of the included studies. The majority of studies were conducted in Switzerland, Israel, Iran, and Belgium. This outcome likely reflects the current availability and accessibility of research on work-related factors affecting mental health service workers in the databases consulted, rather than a targeted selection by the review team. Our search strategy was systematic and comprehensive, with no geographic restrictions, and thus the distribution represents the studies that met the eligibility criteria at the time of the search.
Comment 5:
The authors chose to conduct a scoping review rather than a systematic review. They need to state the differences between these two methods and explain to readers why a scoping review was conducted. In this case, readers would not be able to distinguish differences in methodology, procedures, and results if a systematic review method had been used.
Response 5:
We thank the reviewer for this important observation. We chose to conduct a scoping review rather than a systematic review because our primary aim was to map the available literature, summarize key findings, and identify knowledge gaps regarding work-related factors and their consequences for mental health service workers. Unlike systematic reviews, which focus on a specific research question and often assess intervention effectiveness, scoping reviews allow for a broader exploration of the evidence, including different study designs, populations, and outcomes. Therefore, a scoping review was the most appropriate approach to provide a comprehensive overview of the field, highlight trends, and identify areas requiring further research. In addition, we revised the manuscript in the Methods section (lines 96–103) to clarify these differences and justify the choice of the scoping review approach.
Comment 6:
For a scoping review, there should be a clear identification of gaps, that is, areas with little or no evidence. The authors have highlighted this briefly. We would be pleased to see a more in-depth discussion on this point.
Response 6:
We appreciate the reviewer’s comment regarding the identification of gaps in the literature. We revised the manuscript in the Discussion section (lines 339–340). In response, we have expanded the discussion to provide a more detailed analysis of areas with limited or absent evidence. Specifically, we highlighted gaps concerning the long-term psychological effects of work-related stress, differences across mental health service settings, organizational culture, team dynamics, and the effectiveness of interventions to mitigate worker illness. These additions clarify areas that require further research and provide guidance for future studies aimed at improving occupational health and service quality in mental health settings.
Comment 7:
Could you quantify how many studies were extracted from academic literature, such as peer-reviewed journals, and how many from gray literature? I would like to know the proportion and relative weight of the information from different sources.
Response 7:
Thank you for this important observation. In the final sample of 28 studies, 27 were published in peer-reviewed journals, while 1 study was identified from gray literature sources (retrieved through the Brazilian Digital Library of Theses and Dissertations – BDTD). Thus, the information synthesized in this review is predominantly derived from peer-reviewed academic publications (96.4%), with gray literature accounting for only 3.6% of the included studies.
Reviewer 2 Report
Comments and Suggestions for Authors
Publication is appropriate in the sense that significant issues are present in the field of medicine. Particularly in the field of behavioral health. The authors tend to focus on local issues related to health care in the Brazilian system. The majority of the reviewed publications are from that country. It is unclear why the majority of publications are from Brazil, with a peppering of publications from Switzerland, Israel, Iran, and Belgium. It would be helpful to understand the logic behind this selection process.
The study points out that there are various factors that influence the well-being of mental health workers. It would be very helpful if the authors were able to rank in order particular variables that contribute to untoward consequences of working in the field of mental health. Discussion of key elements would be very appropriate. The authors state that “ The main challenges identified include work overload, excessive working hours, inadequate physical infrastructure, insufficient supplies, understaffing, lack of managerial support, and exposure to physical, biological, chemical, and ergonomic risks.” Are these ordered in terms of prevalence or significance?
The suggestion is to identify which variables dominate the studies and perhaps focus on those, which would be quite helpful to enhance the quality and focus on the most important variables. For example, is the lack of education, work security, or supplies equally important? It is very important to understand which are the dominant factors under what circumstances. It is assumed that perhaps some of those variables could be extracted from the reviewed articles.
The distinction between workers and professionals is not clear. In the Brazilian studies, the word worker was used as a descriptor, unlike in other studies. “The term 'worker' is broader and includes anyone engaged in labor activity, regardless of their level of specialization or formality”. Are professional individuals those who are licensed to provide clinical services? versus workers who are not providing direct clinical care, such as security personnel, what is the difference in educational background between the 2 groups? The educational level may play a role in how an individual manages stress. Comments?
It is important to include all participants who engage in healthcare provisions; however, it would be helpful to separate the groups and point out the sensitivity to stressors. Is it possible to extract this data from the articles used?
It would be helpful in the diagram to identify primary, secondary, and tertiary stressors and their role in particular outcomes. Furthermore, it is important to identify the prominent consequences/outcomes resulting from given stressors. For example, in the section “work organization and process,” is training a significant variable in determining irritability, insecurity, dissatisfaction, and emotion? Can you generate a rank order? It is difficult to imagine that all those factors are equally important.
The pandemic data are included in this section. However, it is not clear how that plays a role in the overall theme, since this was a transitional time. From which the system presumably recovered. It would be important to understand the uniqueness of the situation relative to standard that exists outside of this stressful time.
It is the opinion of this reviewer that this work would have a greater impact if one were able to identify key variables and potentially list other variables as also important but not as critical.
It is not clear what the authors are trying to communicate with the statement “Work must be problematized as a space that shapes relationships and generates either pleasure or suffering in this context.” Clarification would be appreciated.
Author Response
Comment 1
The publication is appropriate in the sense that significant issues are present in the field of medicine, particularly in the area of behavioral health. The authors tend to focus on local issues related to health care in the Brazilian system. Most of the reviewed publications are from that country. It is not clear why the majority of publications come from Brazil, with a dispersion of publications from Switzerland, Israel, Iran, and Belgium. It would be helpful to understand the logic behind this selection process.
Response 1
We appreciate the reviewer’s observation. We revised the manuscript in the Results section (lines 190–195). We have now clarified in the manuscript why the majority of the included studies come from Brazil. As revised, the text explains that this predominance is related to the country’s significant scientific output in the field of mental health, especially following the Psychiatric Reform, which stimulated extensive research on working conditions and professionals’ health in community-based services. International publications were also included, although in smaller numbers, given the specificity of the topic within the Brazilian context.
Comment 2
The study indicates that several factors influence the well-being of mental health professionals. It would be very useful if the authors could classify in order the specific variables that contribute to the adverse consequences of working in mental health. Discussion of the key elements would be highly appropriate. The authors state that “the main challenges identified include work overload, excessive working hours, inadequate physical infrastructure, insufficient supplies, lack of staff, lack of managerial support, and exposure to physical, biological, chemical, and ergonomic risks.” Are these challenges ordered in terms of prevalence or significance? The suggestion is to identify which variables dominate the studies and perhaps focus on them, which would be quite useful to enhance quality and emphasize the most important variables. For example, are lack of education, workplace safety, or supplies equally important? It is very important to understand which factors are dominant and under what circumstances. Presumably, some of these variables could be extracted from the reviewed articles.
Response 2
We thank the reviewer for this important suggestion. We revised the manuscript in the Discussion section (lines 326–337). We agree that classifying the stressors helps provide a clearer understanding of their relative importance. However, given the methodological scope of this study, we did not perform statistical weighting or quantitative synthesis of prevalence or incidence rates, as such analyses are beyond the purpose of a scoping review. Instead, we addressed the reviewer’s concern by organizing the identified stressors according to their recurrence and emphasis across the included studies. This frequency-based categorization, now included in the Discussion section, distinguishes primary, secondary, and tertiary stressors, thereby improving interpretability while respecting the methodological boundaries of a scoping review.
Comment 3
The distinction between workers and professionals is not clear. In Brazilian studies, the word trabalhador (worker) was used as a descriptor, differently from other studies. “The term ‘worker’ is broader and includes anyone engaged in labor activity, regardless of their level of expertise or formality.” Professionals are those licensed to provide clinical services? Compared to workers not providing direct clinical care, such as security staff, what is the difference in educational training between the two groups? Educational level may influence how an individual manages stress. Comments?
It is important to include all participants working in the field of health; however, it would be useful to separate groups and highlight sensitivity to stress factors. Is it possible to extract these data from the articles used?
Response 3
We appreciate this insightful observation. We revised the manuscript in the Discussion section (lines 338–348). Indeed, we noted a conceptual divergence in the terminology adopted by the included studies. In the Brazilian context, the term worker (trabalhador) was employed as a broad descriptor, encompassing all individuals working in mental health services, regardless of their formal training or clinical role. By contrast, studies from other countries tended to use professional to specifically designate licensed practitioners providing direct clinical care. We acknowledge that distinguishing between these groups could offer a more nuanced understanding of differential vulnerabilities to occupational stress. However, the data reported in the included studies were not sufficiently detailed or standardized to enable a stratified analysis by role, professional status, or educational level. For this reason, we addressed the terminology variation qualitatively in the Discussion and highlighted the lack of differentiation as a limitation of the current evidence base. We also agree with the reviewer that this represents a valuable direction for future research, as the level of professional training and direct clinical involvement may significantly shape stress responses.
Comment 4
It would be useful, in the diagram, to identify primary, secondary, and tertiary stressors and their role in specific outcomes. In addition, it is important to identify the prominent consequences/outcomes resulting from certain stressors. For example, in the section “organization and work process,” is training a significant variable in determining irritability, insecurity, dissatisfaction, and emotion? Can you generate a ranking order? It is difficult to imagine that all these factors are equally important.
Pandemic-related data are included in this section. However, it is not clear how this plays a role in the overall theme, since this was a transitional period from which the system presumably recovered. It would be important to understand the uniqueness of the situation in relation to existing patterns outside this stressful period.
In this reviewer’s opinion, this work would have a greater impact if it were possible to identify key variables and potentially list other variables as also important, but not as critical.
It is not clear what the authors are trying to communicate with the statement “Work should be problematized as a space that shapes relationships and generates pleasure or suffering in this context.” Clarification would be appreciated.
Response 4
We appreciate the reviewer’s insightful comments. In line with the suggestion, we revised the Discussion to clarify the relationship between stressors and specific consequences when data permitted. However, as this is a scoping review, the available studies did not allow for a quantitative ranking of these associations. We therefore categorized stressors by frequency of citation rather than impact magnitude, to remain methodologically consistent.
Regarding the COVID-19 pandemic, we now highlight that it represented a transitional and exceptional period, during which existing stressors (e.g., workload, lack of resources) were intensified. We explicitly note that these findings cannot be generalized beyond the pandemic context.
Finally, we rephrased the statement “Work should be problematized as a space that shapes relationships and generates pleasure or suffering” to improve clarity. The revised version reads: “The workplace in mental health services can simultaneously function as a source of professional fulfillment and as a context for stress and suffering, depending on organizational and relational dynamics.”
Round 2
Reviewer 2 Report
Comments and Suggestions for Authors
Thank you for your responses and clarifying remarks.